# New Insights into Hormonal Therapies in Uterine Sarcomas

**DOI:** 10.3390/cancers14040921

**Published:** 2022-02-12

**Authors:** Elena Maccaroni, Valentina Lunerti, Veronica Agostinelli, Riccardo Giampieri, Laura Zepponi, Alessandra Pagliacci, Rossana Berardi

**Affiliations:** 1Department of Oncology, Azienda Ospedaliero-Universitaria Ospedali Riuniti di Ancona, 60126 Ancona, Italy; laura.zepponi@ospedaliriuniti.marche.it (L.Z.); alessandra.pagliacci@ospedaliriuniti.marche.it (A.P.); 2Department of Oncology, Università Politecnica delle Marche, 60126 Ancona, Italy; S1102249@pm.univpm.it (V.L.); veroagostinelli@gmail.com (V.A.); 3Department of Oncology, Università Politecnica delle Marche, Azienda Ospedaliero-Universitaria Ospedali Riuniti di Ancona, 60126 Ancona, Italy; r.giampieri@staff.univpm.it (R.G.); r.berardi@staff.univpm.it (R.B.)

**Keywords:** uterine sarcomas, hormonal therapy, gynaecological cancers

## Abstract

**Simple Summary:**

Uterine sarcomas are rare mesenchymal malignant cancers, and surgery represents the mainstay of treatment for early-stage disease. In metastatic setting, uterine sarcomas’ treatment includes palliative surgery, a metastases resection, chemotherapy and targeted therapy. Hormonal therapies may also represent an effective option. Frequently, ER and PR are highly expressed in uterine sarcomas patients and they represent a favourable prognostic factor associated with improved overall survival. The scope of the present review is to report the existing evidence and future perspectives on hormonal therapy in uterine sarcomas, with a special focus on aromatase inhibitors, progestins and gonadotropin-releasing hormone analogues, in order to clarify their potential role in daily clinical practice.

**Abstract:**

Uterine sarcoma (US) is a rare mesenchymal malignant cancer type, accounting for 3–7% of uterine malignancies. US prognosis is still poor due to high local and distant recurrence rates. As for molecular features, US may present variable oestrogen receptor (ER) and progesterone receptor (PR) expressions, mostly depending on histotype and grading. Surgery represents the mainstay of treatment for early-stage disease, while the role of adjuvant chemotherapy or local radiotherapy is still debated and defined on the basis of histotype, tumour grading and stage. In metastatic setting, uterine sarcomas’ treatment includes palliative surgery, a metastases resection, chemotherapy, hormonal therapy and targeted therapy. As for the chemotherapy regimen used, drugs that are considered most effective are doxorubicin (combined with ifosfamide or alone), gemcitabine combined with docetaxel and, more recently, trabectedin or pazopanib. Hormonal therapies, including aromatase inhibitors (AIs), progestins and gonadotropin-releasing hormone analogues (GnRH-a) may also represent an effective option, in particular for low-grade endometrial stromal sarcoma (LGESS), due to their favourable toxicity profile and patients’ compliance, while their role is still under investigation in uterine leiomyosarcoma (uLMS), high-grade endometrial stromal sarcoma (HGESS), undifferentiated uterine sarcoma (USS) and other rarer US. The present review aims to analyse the existing evidence and future perspectives on hormonal therapies in US, in order to clarify their potential role in daily clinical practice.

## 1. Introduction

Uterine sarcomas (US) are rare mesenchymal malignant tumours, accounting for about 3–7% of all uterine malignancies and less than 1% of all malignancies from the female genital tract [1].

According to the updated World Health Organization (WHO, fifth edition) classification, they can be distinguished based on their histology as uterine leiomyosarcoma (uLMS), low-grade endometrial stromal sarcoma (LGESS), high-grade endometrial stromal sarcoma (HGESS), undifferentiated uterine sarcoma (UUS), adenosarcoma, rhabdomyosarcoma and perivascular epithelioid cell tumour (PEComa). Among them, uLMS is the most common, whereas adenosarcoma, rhabdomyosarcoma, and perivascular epithelioid cell tumour (PEComa) are extremely rare. Carcinosarcomas (malignant mixed müllerian tumours, MMMTs) are no longer considered as sarcomas, as they are dedifferentiated carcinomas with both epithelial and stromal components; nowadays, they are considered, staged and treated as high-grade endometrial cancers [2].

US clinical presentation is non-specific and includes abnormal vaginal bleeding, uterine enlargement, palpable pelvic mass and pelvic pain. Occasionally, symptoms and signs related to tumour rupture (hemoperitoneum), extrauterine extension or metastases might be the first sign of the disease. Distant metastases may occur even at an early stage, and distant relapse after radical surgery is common.

Due to US rarity and histopathological differences, there is a lack of general consensus concerning optimal treatment regimens; furthermore, the majority of published studies focused on US treatment have grouped all kinds of US together and have not considered their relevant differences in clinical behaviour, histopathology and molecular biology. Regarding molecular features, US may present a variable oestrogen receptor (ER) and progesterone receptor (PR) expression, mostly depending on the histotype. In 2016, a study conducted by Davidson et al. on 291 US patients showed that ER and PR were expressed in 53% and 67% of LGESS, 45% and 65% of uLMS, 23% and 31% of HGESS and 47% and 63% of all US, respectively. In this study, a higher objective response rate and prolonged survival were associated with positive expression of PR in patients with stage I LMS [3]. Among all types of US, the highest frequency of ER and PR expression was detected in LGESS. In particular, LGESS usually presents a high expression of ER and PR, related to less aggressive clinical behaviour and a favourable prognosis [4]. Unfortunately, despite significant improvements in surgical techniques and the introduction of innovative treatment options, the US 5-year survival rate remains poor, ranging from 31% to 64% [5].

Radical surgery represents the mainstay of treatment for early-stage disease. The preferred surgical treatment is en bloc total hysterectomy with or without bilateral salpingo-oophorectomy. Lymphadenectomy is not indicated in patients without macroscopically involved lymph nodes and should not be recommended unless the patient has evident extrauterine involvement or clinically suspicious enlarged nodes [6,7,8,9]. The Federation of Gynecology and Obstetrics (FIGO) staging should be used to stage US [7].

The role of adjuvant chemotherapy or local radiotherapy is still controversial and should be offered on the basis of histopathological subtype and stage. In patients with uLMS, adjuvant chemotherapy can be suggested in selected cases with high-risk factors (such as morcellation, tumour size greater than 5 cm, high mitotic index), after accurate discussion with the patient; on the other hand, in patients with early LGESS, observation is the recommended management after surgery; however, it should be noted that some retrospective studies [8] reported that adjuvant hormonal therapy with progestins may determine recurrence risk reduction in these latter patients. Adjuvant treatment for stage I patients is not recommended for the less common histologies HGESS and UUS, while patients with FIGO stage II and III should be offered systemic adjuvant chemotherapy [8].

In patients with metastatic or recurrent disease, systemic chemotherapy represents the standard approach for the treatment of uLMS, HGESS, UUS; first-line treatment is based on anthracyclines either administered as single agents or in combination with ifosfamide or dacarbazine. Another effective combination is represented by gemcitabine-doxetaxel, while trabectedin, pazopanib, dacarbazine and eribulin might also be active. Particularly in uLMS, the ER/PR expression rate in LMS is lower than LGESS, but even uLMS with a positive receptor expression (ER+/PR+) may be sensitive to hormonal therapy.

Owing to the rarity of this kind of tumours, prospective randomised controlled trials of hormonal therapy in US are lacking; however, a few reports (summarised in Table 1 and Table 2) were published on this matter, since the favourable toxicity profiles of these tumours as compared to chemotherapy, coupled with a simple administration and good compliance, have spurred further research on this field.

The present review aims at analysing the existing evidence and future perspectives on hormonal therapies in US, in order to clarify their potential role in daily clinical practice.

## 2. Materials and Methods

The studies included in the present review were selected from PUBMED/GoogleScholar/ScienceDirect databases. We used the terms “Uterine Sarcoma”, “Gynecological Sarcoma”, “Uterine Leiomyosarcoma”, “Endometrial Stromal Sarcoma”, “Uterine Undifferentiated Sarcoma”, for tumour location and the terms “Hormonal Therapy”, “Hormonal Treatment”, “Progestins”, “Aromatase Inhibitors” to select papers focused on hormonal treatment in US.

An article search was conducted independently by three of the co-authors (E.M., V.L. and V.A.). The following inclusion criteria were used to select the studies included in the present review:Type of published study: clinical trial, review articles, guidelines;Selected population: patients affected by uterine sarcomas both in early-stage or advanced disease;Evaluation of ER and/or PR expression;No limitation regarding the year of the publication of papers was used.

Conversely, exclusion criteria are reported below:Meeting or conference abstracts only;Duplicate papers;Articles not written in English language.

## 3. Results

### 3.1. Hormonal Therapy in Endometrial Stromal Sarcomas

Endometrial stromal sarcomas (ESS) are rare mesenchymal malignant tumours, representing 0.2–1% of all uterine cancers and 10–15% of all uterine sarcomas [10]. There are two types of ESS, based on cell morphology and mitotic rate: LGESS and HGESS [10,11]. LGESS are usually slow-growth malignancies with a penchant for late recurrence. HGESS are more aggressive than LGESS, characterised by a worse prognosis [12]. LGESS have a better prognosis than HGESS and are more common [13].

Due to the rarity of these diseases, the lack of large-scale studies makes it difficult to clearly define the best specific treatment and most appropriate sequence of therapies [8].

#### 3.1.1. Low Grade ESS

Total hysterectomy (TH) and bilateral salpingo-oophorectomy (BSO) are the current standards of treatment [14]. Systematic lymphadenectomy (LAD) remains controversial because it does not have a great impact on survival. In their meta-analysis Si et al. (2017) report that LAD may not be recommended in patients with LGESS unless the patient has an advanced disease or suspicious lymph nodes [15]. The preservation of fertility can be considered for young nulliparous women at an early stage [14]. Since the benefit of BSO compared with other forms of surgical resection needs to be defined for LGESS, ovaries preservation is possible for young patients with stage I low-grade ESS, as it does not increase recurrence risk [16,17].

Chemotherapy and radiotherapy can represent possible adjuvant treatments choices in advanced LGESS, although there is a lack of solid evidence suggesting a benefit of prognosis. On the other hand, hormonal therapy has a relevant role in the treatment of LGESS, as ER and PR are expressed in 70% and 90% of cases, respectively. For this reason, hormonal growth signals play a critical role in the development of this group of tumours [8].

Currently, observation is the suggested management for stage I patients after surgery, particularly in postmenopausal women [8,18].

For more advanced stages, hormonal therapy contributes to a recurrence risk reduction when used as an adjuvant therapy after surgery [12]. Furthermore, several case reports and retrospective studies reported that hormonal therapy is more effective for recurrent patients compared to radiotherapy and chemotherapy [8].

However, all of these studies are limited by the small number of patients taken into account. For this reason, the role of endocrine treatment is not well defined and remains controversial.

##### Progestins in LGESS

Progestins are synthetic derivatives of progesterone. As PR is highly expressed in LGESS, progestins might have a relevant role in uterine sarcomas’ treatment [19]. Progestins bind to PR, determining an antioestrogenic activity and a reduction in stromal endometrial proliferation [20]. The antioestrogenic activity works in several different ways, determining the reduction in circulating oestrogens [21].

Progestins are the most commonly used first-line hormonal therapy in LGESS because they can determine a relatively high response rate [22]. Based on the literature, progestins were used in LGESS, either as an adjuvant treatment after surgery or in the treatment for recurrent and metastatic disease [22]. Megestrol acetate (MA) and medroxyprogesterone acetate (MPA) are the most-used progestins. However, mifepristone, a selective progesterone receptor modulator, has also been used because of its antiprogestational activity [20,23,24].

Due to the rarity of this group of tumours, only a few case reports and retrospective studies with small numbers of patients have been published on this matter; there is no global consensus about post-operative treatments for these patients. Among the retrospective studies that are available regarding progestins in adjuvant settings (Katz et al., 1987 [25]; Chu et al., 2003 [26]; Leath et al., 2007 [12]; Beck et al., 2012 [27]; Malouf et al., 2010 [28]; Cheng et al., 2011 [18]), they show lower recurrence rates in all stages for patients receiving those drugs.

Beck et al. [27] reported that patients in early-stage disease receiving continuous progestin therapy after surgery have a lower recurrence rate than those who did not receive any treatment after surgery (14.3% vs. 38.5%, *p* = 0.26), as well as in all stages of disease (33% vs. 50%, *p* = 0.38).

Contrastingly, in a large retrospective cohort analysis including 2414 LGESS patients, Seagle et al. [13] reported that adjuvant hormonal therapy had little benefit on OS. It should be mentioned, however, that only 12.7% (307 patients) LGESS patients had actually received hormonal therapy. Moreover, a small number of women in the LGESS cohort, may have actually had HGESS. These two factors might have contributed to the disappointing results of this study.

In recurrent or metastatic LGESS settings, there are several case reports and retrospective studies in the literature: in several case series of metastatic LGESS treated with progestin (Ioffe et al., 2009 [29]; Pink et al., 2006 [30]; Chu et al., 2003 [26]; Dahhan et al., 2009 [31]; Yamazaki et al., 2015 [32]; Mizuno et al., 2012 [33]), authors reported relatively high response rates and prolonged times to progression.

Ioffe et al. [29] reported that hormonal therapy with progestins produces a clinical response in patients with metastatic ESS, determining stable disease or partial response. This study also evaluated the prognostic significance of oestrogen receptor expression in ESS, demonstrating improved overall survival in ER-positive uterine sarcomas compared with ER-negative patients (median OS 36 vs. 16 months, *p* = 0.004). They suggested that ER status could be considered as a prognostic marker and a therapeutic target in uterine sarcomas.

Pink et al. [30] reported the results of a retrospective analysis comprising a few patients treated with first-line MPA, where disease control was achieved in most cases. Similarly, Chu et al. [26] reported that recurrent patients treated with progestin therapy had complete responses from 4 to 15 years [26]. Moreover, Cheng et al. [18] reported that hormonal therapy was more effective than radiotherapy and chemotherapy for patients with recurrent LGESS, with a 27% overall response rate and 53% stable disease rate. The median TTP was 24 months.

All these studies included patients with ER-/PR-positive ESS who, when treated with progestins, in most cases, responded with a stability of their tumour, with a total effective rate of about 86.9% [22]. However, cases of PDs during treatment with progestins were also reported, perhaps due to PR absence of expression or other mechanisms determining progestin resistance [20,26].

In this regard, Maenohara et al. [34] recently described a case report of dydrogesterone- and MPA-resistant metastatic patients, who were successfully and safely treated with dienogest (DNG), suggesting that DNG treatment may control those tumours that were resistant to other hormonal therapies.

Another important concern is the safety profile of progestins: although progestins are usually recommended as adjuvant therapies, it is very difficult to keep patients on prolonged therapy with them, due to the side effects. For this reason, sometimes, the use of aromatase inhibitors is preferred as they have a more favourable side effect profile compared to progestins [24].

##### Aromatase Inhibitors in LGESS

Aromatase inhibitors (AIs) suppress oestrogen biosynthesis by blocking aromatase activity [35]. Aromatase is the enzyme that catalyses oestrogen biosynthesis from androgens, and it is determined by gene CYP19 [35]. Intra-tumoural aromatase expression has been reported in approximately 80% of ESS [36].

AIs have been traditionally considered the second-line hormonal therapy in LGESS, after progestins’ treatment failure [22]. First-generation AIs (Aminoglutethimide) and second-generation AIs (Formestan, Fadrozol) had serious side effects related to both mineral and glucocorticoid synthesis inhibition. On the other hand, third-generation AIs (Letrozole, Anastrozole and Exemestane) have an acceptable tolerance profile and can be administered orally [37]. Moreover, they have shown a more favourable efficacy profile over MA, with minor side effects and a higher therapeutic index compared with progestins [20,31]. For this reason, third-generation AIs were used empirically as first-line hormonal treatment [22]. For patients with ESS, the same dosage as for breast cancer (1 mg/day Anastrozole, 2.5 mg/day Letrozole, 25 mg/day Exemestane) was adopted as off-label treatment for these patients.

There are several published studies about AIs in LGESS and they seem to have a nearly equivalent or even better response as a first-line therapy than as a second-line therapy after the failure of progestins; responses were usually partial, but complete responses were also reported.

Altal et al. [38] described the complete remission of advanced low-grade endometrial stromal sarcoma with extensive metastasis to the lung, bladder, ureteral orifice and iliac lymph nodes after surgery and hormonal therapy with Letrozole 2.5 mg daily.

Similarly, Spano et al. [39] reported complete responses in two patients with advanced ER- and PR-positive ESS after surgery and hormonal therapy with Aminoglutethimide, a first-generation AI. In particular, one of the patients had a complete remission for 14 years with aminoglutethimide; the other patient, after having diarrhoea and asthenia as side effects, switched instead to Letrozole and achieved CR for 7 years.

Case series and retrospective studies reported treatment with AIs used either in first- or in second-line treatments. In regard to first-line treatment, Leunen et al. [40] described an excellent response in a patient with recurrent low-grade ESS after therapy with letrozole. Alkasi et al. [41] described the long-term survival of a 28-year-old patient with metastatic ESS treated with Goserelin and Anastrozole after surgical intervention. Similarly, Pink et al., 2006 [30]; Ioffe et al., 2009 [29]; Dahhan et al., 2009 [31]; Ryu et al., 2015 [42]; and Yamaguchi et al., 2015 [43] described the effectiveness of first-line AIs in the setting of recurrent or metastatic LGESS.

Most of the previous studies (Spano et al., 2003 [39]; Pink et al., 2006 [30]; Ioffe et al., 2009 [29]; Dahhan et al., 2009 [31]; Ryu et al., 2015 [42]) also described cases when AIs were used as second-line therapy, after surgery and progestins, reporting many cases of complete or partial response, or stable disease. Cases of PD are also described, but these are fewer compared to responses.

A more recent phase II, prospective, single-arm, open-label trial by Friedlander et al. aimed to evaluate the activity of anastrozole 1 mg/day in recurrent or metastatic LGESS patients with ER ± PR + expression. LGESS patients with measurable disease were included and treated until progressive disease or unacceptable toxicity was observed. The primary endpoint was the clinical benefit rate (CBR), comprising complete/partial response at 3 months. Fifteen LGESS patients were included, and CBR at 3 months was 73% (also resistant at 6 months, with a ORR of 26.7%). Median PFS was not reached. The authors concluded that the ORR with anastrozole was lower than reported in retrospective series, even if the CBR was high and resistant [44].

Ioffe et al. [29] suggested that AIs, due to their favourable side effects profile compared to progestins, as well as potential survival benefits, should be considered either as first-line or second-line treatments for LGESS. In particular, some studies reported that AIs were superior to progestins for recurrent ESS. [30,45]. The latest National Comprehensive Cancer Network (NCCN) guidelines recommend AIs as preferred therapeutic regimens for LGESS [46].

##### GnRh-Analogues in LGESS

Gonadotropin-releasing hormone analogues (GnRh-a) (Leuprolide, Goserelin, and Triptorelin) in premenopausal women, inhibit the pituitary ovarian axis, determining the suppression of ovarian oestrogens to a level equivalent to the postmenopausal status [20,37]. Moreover, immunohistochemical studies reported that the GnRh receptor (GnRh-R) was expressed in most cases of ESS; this fact suggested an additional mechanism of action of GnRh-a [47,48].

Some cases are reported in the literature regarding the use of GnRh-a in different settings, either as monotherapy or in combinations with progestins or AIs. Mesia et al. [49] described the reduction in tumour size in a patient with stage I ESS, after being preoperatively treated with 3.75 mg/month of Leuprolide acetate, for two months. Burke et al. [50] reported a case of recurrent ESS treated with Triptorelin, achieving the control of the progression and the reduction in tumour size. Another case report by Scribner et al. [51] described the reduction in the tumour size of an inoperable LGESS treated with Leuprolide acetate in combination with MA.

The limited number of studies reported in the literature makes it difficult to establish the efficacy of GnRh-a, and because of that, further studies are needed to determine their role in LGESS [8].

##### Oestrogen Replacement Therapy and Tamoxifen in LGESS

Based on the literature, oestrogen replacement therapy (ERT) and SERMs seem to have a negative effect on patients with ESS [30]. In fact, the incidence of uterine sarcomas is higher in patients receiving hormone replacement regimens containing oestrogens or tamoxifen [8,52].

Moreover, Tamoxifen and ERT appear to be contraindicated in patients diagnosed with LGESS, as confirmed by standard guidelines [26]. In this regard, Pink et al. [30] reported that patients affected by ESS should not be treated with ERT or Tamoxifene and, should the patient be on incidental therapy with these drugs, withdrawal of these treatments is suggested, as it can result in tumour stabilization in some cases.

SERMs (Tamoxifen, Toremifene and Fulvestrant), perform their action by binding to the oestrogen receptor [22]; in breast tissue, Tamoxifen acts as an ER antagonist, inhibiting the activity of oestrogens and, for this reason, it is commonly used as to treat breast cancer. On the other hand, in the uterus, Tamoxifen seems to have a pro-oestrogenic effect, promoting disease evolution [22]. The opposite effect of SERMs in the breast and uterus may be due to the different expressions of co-regulatory proteins [22,31].

#### 3.1.2. High-Grade ESS

HGESS has a higher rate of recurrence and poorer prognosis than LGESS. Moreover, it is still controversial whether adjuvant therapies after surgery can improve survival compared with observation [8]. In a large observational retrospective cohort analysis, Seagle et al. reported that increased tumour size, distant or nodal metastasis, no lymphadenectomy at surgery and positive surgical margins were negative prognostic factors for HGESS [13].

Systemic therapy or RT can be offered to patients with stage II-III disease, while for stage I, patient observation is the main strategy [8]. HGESS are usually ER- and PR-negative: in this regard, hormonal therapy is not generally considered as an option of treatment [29]. Another topic to analyse regarding molecular subtypes is HGESS. In fact, in one study, Momeni-Boroujeni et al. reported that, while ER and PR expression may be seen in some HGESS, the low ESR1 RNA expression may determine the limited efficacy of endocrine therapy in these tumours. In this regard, two patients with ER-/PR-positive but ESR1-downregulated stage I HGESS were treated with hormone therapy and recurred at 12 and 36 months after primary resection [53].

Few data are available in the literature and, for this reason, the choice of chemotherapy or other treatments is related to the evaluation of risk/benefit assessment and performance status [8,54].

### 3.2. Hormonal Therapy in Uterine Leiomyosarcoma

uLMS is the most-common uterine sarcoma, accounting for about 60% of all uterine sarcomas [55]. It has a very aggressive behaviour, with a poor prognosis even if confined within the uterus [56]: the recurrence rate is high (50–70%) and a 5-year OS is less than 50% [10,57].

Several studies analysing the immunohistochemistry characteristics of uLMS showed that nearly 50% of them express oestrogen and progesterone receptors: uLMS 25–60% of cases are ER-positive and 35–60% are PR-positive, respectively [58,59]. Moreover, in a study by Ioffe et al. including 54 women affected by uterine sarcomas, it was found that among 13 patients (24%) affected by uLMS, 100% of them were ER-positive [29].

Hormonal receptors may also have a significant prognostic value: their expression correlates with better OS and PFS in most studies, particularly in patients treated with hormonal therapy [60,61]. In addition to that, an increasing amount of evidence has suggested a predictive value of ER and/or PR expression; in this setting, hormonal treatments are related to greater response or prolonged disease stability [26,39].

According to these data, oestrogen manipulation seems to have a potentially active role in uLMS disease [62].

In uLMS, hormonal therapy has been investigated both in adjuvant and metastatic settings or at time of recurrence, despite having a biological rationale based more on their impact on hormone-sensitive breast cancer rather than proper data on the less-common endometrial cancer [20,39,63,64].

To date, only a few case reports, short retrospective studies and one prospective single-arm phase II clinical trial of hormone-positive uLMS have been published [29,65,66]. Conversely, no randomised trials of hormone therapy have been conducted in uLMS, due to the aggressive clinical behaviour and rarity of the disease [37].

A retrospective study by Ioffe et al., published in 2009, was conducted in order to clarify the prognostic role of ER expression in uterine sarcoma, and to verify the hypothesis that ER-positive uterine sarcomas could respond to hormonal therapy. Survival analysis showed that patients with ER-positive US had improved OS when compared with ER-negative patients (median OS 36 vs. 16 months, *p* = 0.004). In a multivariate analysis, ER positivity maintained its significance as an independent predictor of survival (HR = 0.32, CI 0.12–0.89, *p* = 0.03). Patients with stage I–II LGESS were excluded from the survival analysis, due to their significantly better prognosis.

Regarding hormonal treatment, 4 patients were treated in the adjuvant setting and maintained the prolonged remission of disease (range of follow up: 18–68 months), while 18 patients received hormonal therapy in a recurrent or progressive disease setting; 14 (78%) obtained stable disease or complete or partial response (range of follow up: 6–124 months). The authors concluded that ER expression is common in US, and it is associated with improved overall survival. On this basis, hormone therapy should be considered in patients with primary and recurrent ER-positive uterine sarcomas [29].

In patients with early-stage disease, another small, randomised, open-label phase II trial aimed to evaluate the efficacy of letrozole in patients with newly diagnosed early-stage uLMS with positive hormonal receptors expression. The primary endpoint of this study was the recurrence rate reduction in patients treated with letrozole. Nine patients were enrolled: four patients were in the experimental arm and five patients were in the observation arm. The median PFS for the experimental arm was not reached (NR), compared to 17.3 months in the control arm. The progression-free rate at 12 and 24 months was 100% for patients receiving letrozole, compared to 80% at 12 months and 40% at 24 months for patients in the observation arm. Due to early study closure, the authors stated that these promising observations require further investigation [66].

Regarding patients with advanced uLMS, a small retrospective trial was performed including 16 uLMS patients expressing ER and/or PR in 100% of cases and treated with AIs as a first-line treatment. A median PFS in first-line treatment of 14 months was observed, especially in patients with low-grade compared to high-grade uLMS (20 months vs. 11 months), and in moderately/strongly ER-positive compared to weakly ER-positive uLMS (20 months vs. 12 months); an objective response was observed in 12.5% with a rate of clinical benefit in 62.5% [62]. Furthermore, among the 16 enrolled patients, a second-line treatment with exemestane or anastrozole was used in 6 of them after disease progression, with a 1-year progression free rate of 80% and an objective response observed in 1 patient [62].

Another small single-arm phase II study by George et al. aimed to evaluate the 12-weeks’ progression-free survival (PFS) in advanced/unresectable uLMS patients with an ER- and/or PR-expression-positive status as confirmed by immunohistochemistry, treated with letrozole 2.5 mg/daily. Twenty-seven patients were enrolled, with a median of two prior treatment regimens. The 12-week PFS rate was 50% (90% confidence interval, 30–67%), while the best response achieved was stable disease in 14 patients (54%). Three patients, all expressing ER and PR in >90% of tumour cells, received letrozole for more than 24 weeks; letrozole was also well-tolerated. The authors concluded that letrozole met protocol-defined criteria as an active agent in patients with advanced uLMS, in particular for those whose tumours strongly expressed ER and PR [67].

In contrast with the aforementioned studies, some trials seem to show that some patients might have lower response rate despite high ER and/or PR expressions in the tumours [68]. O’Cearbhaill et al. tried to explain these controversial results by commenting on the fact that the improved outcome of ER-positive tumours could be due to the more favourable biological behaviour, as opposed to the therapeutic effect of endocrine therapy in ER-positive uLMS [69].

In conclusion, further clinical trials are needed in this setting, and hormonal therapy still cannot be routinely recommended in uLMS, despite promising data [70].

#### Type of Hormone Therapy in uLMS

Some studies demonstrated a relationship between the use of tamoxifen or HRT in uterine sarcomas and quicker disease progression [71].

Although tamoxifen is widely used in patients with breast cancer, being a selective modulator of ER, it causes an agonist effect on endometrial stromal cells, and thereby stimulates endometrial tumour growth [37,72]. For these reasons, tamoxifen is contraindicated in patients with ER-positive uLMS.

Nowadays, the role of progesterone and progestin treatment in uLMS is still being discussed: more than ten years ago, two case reports were published regarding the use of progestin in this sarcoma [22]. A woman with lung metastases was treated with medroxyprogesterone acetate after surgery, achieving 12 months of PFS. However, in the following years, no new cases related to the use of progestin were reported [73]. A few trials demonstrated that progestin could be used in metastatic uLMS as a first-line treatment [61].

Some studies, both in metastatic and adjuvant settings, showed that by the use of AIs, better objective response rates and PFS were seen in patients with ER-positive uLMS, compared to those with ER-negative tumours [20]. The strongest data regarding the use of AIs in uLMS come from a retrospective study by O’Cearbhaill et al. [69] on 34 patients (74% received letrozole), demonstrating clinical stability in 32% of patients and a median PFS of 2.9 months.

AIs are administered at the same treatment dose as breast cancer [20]. Based on the available data among AIs, letrozole is primarily used as a first-line hormonal treatment, while exemestane and anastrozole are used as a second-line treatment [22].

In addition, based on standard treatment in breast cancer patients, AIs with GnRh-a combinations should be used in premenopausal women with uLMS [18]. In metastatic uLMS, progestin could also be used as demonstrated in some small trials in first-line settings [61].

### 3.3. Hormonal Therapy in Undifferentiated Endometrial Sarcoma

Undifferentiated endometrial sarcoma is a poorly differentiated sarcoma and ER-positive and/or PR-positive expression is rare [8]. Indeed, hormone therapy is not recommended in these sarcomas [18].

### 3.4. Hormonal Therapy in Aggressive Angiomyxoma

Aggressive angiomyxoma is a hormonal-dependent tumour with both ER and PR expression in 90% of cases [74]. Some reports in the literature underline that GnRh-a could be used as first-line treatment or as adjuvant treatment after surgery [75]. Fuca et al. reported a trial of 36 patients in which 13 patients with locally advanced disease received first-line systemic treatment with hormone therapy, with an overall response rate (ORR) of 8/13 (62%) and a median PFS of 24 months (95%, CI) [76].

All the results are summarised in Table 1 and Table 2.

**Table 1 cancers-14-00921-t001:** Summary of data upon hormonal therapy used for the treatment of ESS.

**Early Stage**						
**Study (Year)**	**Type of Study**	**Number of Patients**	**Setting**	**Treatment**	**Results**	**Response**
Katz et al. (1987) [25]	Case series	2	Adjuvant	MA	NED	Response duration: 24–72 months
Malouf et al. (2010) [28]	Retrospective study	4	Adjuvant	MA	4 NED	NA
**Both early and advanced stage**						
**Study (year)**	**Type of study**	**Number of patients**	**Setting**	**Treatment**	**Results**	**Response**
Chu et al. (2003) [26]	Retrospective study	13–8	Adjuvant–Metastatic	MA/NSP	9 NED/4 recurred–4 CR/3 SD/1 PD	Response duration: 18–180 months
Cheng et al. (2011) [18]	Retrospective study	25–30	Adjuvant–Metastatic	NSP	25 NED–5 CR/3 PR/16 SD/6 PD 10-year PFS rate: 43% 10-year OS rate: 85% ORR: 27%; median TTP: 24 months	10-year PFS rate: 43% 10-year OS rate: 85% ORR: 27%; median TTP: 24 months
**Advanced disease**						
**Study (year)**	**Type of study**	**Number of patients**	**Setting**	**Treatment**	**Results**	**Response**
Friedlander et al., 2019 [44]	Phase II, Open label	15	Advanced/ Metastatic	Anastrozole	1 CR/3 PR/7 SD CBR at 3 months: 73% Median PFS not reached ORR 26.7%	CBR at 3 months: 73% Median PFS not reached ORR 26.7%
Pink et al. (2006) [30]	Case series	8	Advanced/ Metastatic	MPA	1 CR/1 SD/1 PD	Response duration: 9–50 months
				5 Letrozole	4 PR/1 PD	Response duration: 3–37 months
Dahhan et al. (2009) [31]	Retrospective study	11	Advanced/ Metastatic	MA	4 CR/3 PR/1 SD	Response duration: 4–252 months
				3 Letrozole	2 PR/1 PD	
Ioffe et al. (2009) [29]	Retrospective study	8	Advanced/ Metastatic	MA/MPA	1 PR/3 SD/1 PD	Response duration: 6–124 months
				3 Letrozole	2 PR/1 CR	
Mizuno et al. (2012) [33]	Case series	6	Advanced/ Metastatic	MPA	3 PR/3SD	After 6 months of therapy: overall response rate: 50%; disease control rate: 100%.
Yamazaki et al. (2015) [32]	Retrospective study	9	Advanced/ Metastatic	MPA DDG GnRHa	3 CR/2 PR/1 SD/2 PD/1 NA	Response rate: 63.6%. Disease control rate 72.7%
Spano et al. (2003) [39]	Case series	2	Advanced/ Metastatic	2 Aminoglutethimide	2 CR	Response duration: 84–168 months
Altman et al. (2012) [77]	Retrospective study	4	Advanced/ Metastatic	4 Anastrozole (1 switched to Letrozole, 1 to Exemestane)	3 SD/1 PR	NA
Ryu et al. (2015) [42]	Case series	2	Advanced/ Metastatic	2 Letrozole	2 CR	Response duration: 3–124 months
Yamagushi et al. (2015) [43]	Retrospective study	5	Advanced/ Metastatic	5 Letrozole	1 PR/2 CR/2 SD	Time to recurrence (TTR) range: 42–192 months

MA: megestrol acetate; NED: no evidence of disease; CR: complete response; NSP: not specified progestins; PR: partial response; MPA: medroxyprogesterone acetate; DDG: dydrogesterone; GnRHa: gonadotropin-releasing hormone analogues; SD: stable disease; PD: progression of disease; CBR: clinical benefit rate; PFS: progression-free survival; ORR: overall response rate NA: not available.

**Table 2 cancers-14-00921-t002:** Summary of data upon hormonal therapy used for the treatment of uLMS and aggressive angiomyxoma.

**Early Stage**						
**Study (Year)**	**Type of Study**	**Number of Patients**	**Setting**	**Treatment**	**Results**	**Response**
Slomovitz et al. (2019) [66]	Open-label phase II trial	4/9 experimental arm	Early stage	AI	NED (3/4) PD (1/4)	Median PFS: not reached (NR). Progression-free rate at 12 and 24 months: 100%
**Both early and advanced stage**						
**Study (year)**	**Type of study**	**Number of patients**	**Setting**	**Treatment**	**Results**	**Response**
Ioffe et al. (2009) [29]	Retrospective study	3	Adjuvant	AI	NED	Response duration: 30–50 months
		5	Advanced/ Metastatic	AI (4/5) TAM (1/5)	SD (3/4) PR (1/4) SD (1/1)	Response duration: 18–72 months
**Advanced disease**						
**Study (year)**	**Type of study**	**Number of patients**	**Setting**	**Treatment**	**Results**	**Response**
George et al. (2014) [67]	Phase II	27	Advanced/ Metastatic	Letrozole	SD (14/27) PFS at 12 weeks: 50%	PFS at 12 weeks: 50%
Thanopoulou et al. (2014) [62]	Retrospective study	16	Advanced/ Metastatic	1st line AI +/− GhRHa	SD (10/16) PD (4/16) PR (2/16)	Median PFS: 14 months (95% CI: 0–30 months)
				2nd line (6/16) AI +/− GnRHa	PR (1/6) SD (3/6) PD (2/6)	Median PFS: not reached. The 1-year progression-free rate for the 2nd line AI was 80%.
O’ Cearbhaill et al. (2010) [69]	Retrospective study	34	Advanced/ Metastatic	AI	PR (3/34) SD (11/34) PD (20/34)	Median PFS: 2.9 months (95% CI: 1.8–5.1). The 1-year PFS rate was 28% (95% CI: 11–48%).
Kim et al. (2013) [73]	Case report	1	Advanced/ Metastatic	MPA	SD	PFS: 12 months
Fuca et al. (2019) [76]	Retrospective study	36	Advanced/ Metastatic	1st line (13/36) AI +/− GnRHa (11/13) TAM (2/13)	CR (2/13) PR (4/13) SD (5/13) PR (2/2)	Overall response rate (ORR) of 8/13 (62%). Median PFS of 24 months (95%, CI). Median time to response: 3 months. 7 out of 13 patients (53.8%) experienced a progression of disease with a median PFS of 24.6 months
				2nd line (3/13)AI +/− GnRHa	CR (1/3) PR (1/3) SD (1/3)	Time to best response range: 2.8–21.9 months; PFS range: 7.3–79.5 months

MA: megestrol acetate; NED: no evidence of disease; CR: complete response; NSP: not specified progestins; PR: partial response; MPA: medroxyprogesterone acetate; SD: stable disease; AI: aromatase inhibitors; PD: progression of disease; TAM: tamoxifen; GnRHa: gonadotropin-releasing hormone agonist. NA: not available.

## 4. Discussion

Uterine sarcomas frequently present ER and PR expression, with a variable rate of expression mostly depending on histotype and grading. Among all US types, the highest frequency of ER and PR expression was detected in LGESS, but other histotypes, particularly uLMS, may also express hormonal receptors [29].

Moreover, ER and PR expression also represent a favourable prognostic factor associated with improved overall survival in US patients [3], and the co-expression of ER and/or PR with the androgen receptor (AR) is associated with significantly better 5-year DFS and OS, compared to patients with negative AR. AR may be an independent prognostic marker regardless of ER/PR [78].

This common molecular feature has led to the evaluation of endocrine treatments as valid therapeutic options for US both in adjuvant and salvage settings; this was also due to their favourable toxicity profile when compared to chemotherapy and ease of use, as most of them are oral drugs, allowing for long periods of treatment with good patient compliance and negligible side effects.

Unfortunately, US tumours are extremely rare gynaecological malignancies characterised by a different clinical behaviour depending on each affected patient. In addition to this, most studies regarding hormonal treatments efficacy are retrospective analyses, case-reports or case series. There are or only a few phase II trials with a small sample size, including the heterogeneity in patients’ inclusion criteria and different histotypes, drugs and schedules (the dosage of hormonal agents in US was extrapolated from the data on breast cancer patients). These two factors contribute to an unclear scenario where the real effectiveness of hormonal therapies in each US histotype can be difficult to estimate.

Despite the aforementioned limitations, in patients with US treated with endocrine therapies, both objective response rates and improved survival seemed to be associated with the positive expression of the hormonal receptor. This is particularly true in LGEES, as it is characterised by the highest percentage of ER and PR expression among all types of US. The study by Cheng et al. [18], conducted on 74 LGESS patients, showed that endocrine treatment with progestins was even more effective than chemotherapy or radiotherapy in terms of response rate, durable response and disease stabilization, confirming that LGESS is a hormone-dependent malignancy, with hormonal therapy displaying an activity in recurrent disease. Similar results were obtained by Ioffe et al. [29], who reported that hormonal therapy with progestins led to clinical responses in patients with metastatic ESS, determining stable disease or partial response.

Other than progestins, AIs are also active in LGEES patients: there are series of published data that suggest that they might achieve nearly the equivalent or even better response rates than second-line chemotherapy after the failure of progestins [29,38,39]. Data regarding GnRH-a use are still relatively scarce in terms of establishing their efficacy in LGESS patients.

In early-stage disease, no randomised trials are available comparing progestins versus aromatase inhibitors, and even if progestins are considered the most-effective drugs, observation alone remains the recommended management for stage I patients after surgery, especially in postmenopausal women [8]. In recurrent/metastatic disease LGESS, most guidelines recommend hormonal therapy as the standard management, since it induces durable responses or disease stabilization, improving overall survival [8,22,30,33,39,43,49,50]. In LGESS, chemotherapy is administered in hormone-unresponsive or hormone-progressive patients [8].

In contrast with LGESS, HGESS generally have a lower or absent ER and PR expression and are characterised by an extremely poor prognosis: chemotherapy, palliative radiotherapy and surgery have been described in advanced and recurrent disease, while endocrine therapies are not effective [8]. Moreover, recent recognition of HGESS with different genotypes suggests that some tumours previously classified as undifferentiated US may represent a misdiagnosed HGESS [79].

Regarding uLMS, the expression rate of ER/PR is lower than LGESS, but uLMS with an immunohistochemical expression of ER and/or PR expression could also be sensitive to endocrine therapy. Unfortunately, data regarding hormonal therapies in uLMS are even more scarce than in LGESS.

Only a few prospective phase II trials have been published, evaluating the role of adjuvant letrozole in patients with early-stage and positive hormonal receptors in uLMS. [44,67] Among them, Slomovitz et al. [66] reported that in nine patients, 12- and 24-month progression-free rates were both 100% for patients receiving letrozole compared to 80% at 12 months and 40% at 24 months for patients in the observation arm; these promising results are based on a sample that is too limited to be extended to a wider population. Similar considerations should be made for patients with advanced uLMS expressing ER and/or PR treated with letrozole [67].

Other published evidence regarding the possible efficacy of hormonal therapies in uLMS mostly derives from retrospective studies that included only a few patients: in the retrospective study by Ioffe et al. [29] only 13 (24%) out of 54 enrolled patients had an uLMS. Even if in US patients ER expression resulted as a positive prognostic factor and endocrine treatment was associated with prolonged remission, several biases should be considered. In fact, all hormonal therapies were allowed (progestins, AIs or tamoxifen), courses of treatment were determined by physician preference, including combinations of chemotherapy, radiation, and hormone therapy, and both patients in adjuvant settings and with advanced disease were included in the final analysis. As a consequence, no information on the type of drug, setting or treatment duration could be extrapolated [29]. Similarly, the retrospective study by Thanopoulou et al. [62] included only 16 women with uLMS. Even in the enrolled population, the type of treatment and setting were more homogeneous, comprising patients with advanced disease receiving first- or second-line therapy with aromatase inhibitors. This study also showed an increase in PFS and the association with clinical benefit, especially in patients with low-grade uLMS and in moderately/strongly ER-positive tumours. Due to the lack of strong evidence, hormonal therapies are not recommended as a standard treatment in either adjuvant settings and advanced uLMS [8].

To date, the use of targeted therapies for US is still limited [80] and the only approved targeted agent for advanced STSs after the failure of standard chemotherapy is pazopanib, an oral multikinase angiogenesis inhibitor [81]. Moreover, the combination of hormonal therapy with targeted agents could be promising in the treatment of US. In the future, comprehensive genomic profiling may help us to improve treatment by using targeted therapies based on tumour-specific mutational patterns in extremely rare diseases such as US.

## 5. Conclusions

Hormonal therapies represent a useful option, especially in some types of US, such as LGESS, even if clear data on the type of drug, setting or treatment duration are still lacking. In the future, larger retrospective analyses and randomised controlled clinical trials are needed to clarify these issues.

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
