# Peer review of "New Insights into Hormonal Therapies in Uterine Sarcomas"

_cancers, 2022, doi:10.3390/cancers14040921_

Round 1

Reviewer 1 Report

Reply to Authors

The chosen topic is interesting considering the rarity of uterine sarcomas and the difficulty of studying and establishing an adequate hormonal therapy. Indeed, the lack of extensive clinical trials and clear recommendations on the use of hormonal therapy in uterine sarcomas make the subject difficult to approach.

Major revisions

  • Materials and methods – eligible study design, inclusion, and exclusion criteria
  • Line 195 -I suggest adding the total clinical effective rate to quantify the response to treatment according to the article.DOI: 10.1002/cam4.2044
  • Line 250 - (NCCN) guideline recommends aromatase inhibitors as preferred therapeutic regimens for LG-ESS (Abu-Rustum, 2021).
  • Lines 305-307 and lines 324-327 --- repetitive mention
  • Line 464 there is not only one prospective phase II trial –
    • George, S., Feng, Y., Manola, J., Nucci, M.R., Butrynski, J.E., Morgan, J.A., et al., 2014. Phase 2 trial of aromatase inhibition with letrozole in patients with uterine leiomyosarcomas expressing estrogen and/or progesterone receptors. Cancer 120,738–743,
    • Friedlander M, Benson C, O'Connell RL, Reed N, Clamp A, Lord R, Millan D, Nottley S, Amant F, Steer C, Anand A, Mileshkin L, Beale P, Banerjee S, Bradshaw N, Kelly C, Carty K, Divers L, Alexander L, Edmondson R. Phase 2 study of anastrozole in patients with estrogen receptor/progesterone receptor-positive recurrent low-grade endometrial stromal sarcomas: The PARAGON trial (ANZGOG 0903). Gynecol Oncol. 2021 Apr;161(1):160-165. DOI: 10.1016/j.ygyno.2021.02.016. Epub 2021 Feb 16. PMID: 33608144.
  • I would like only to suggest introducing a sentence in the Discussions section - Co-expression of ER and/or PR with androgen receptor (AR) was associated with significantly better 5-year DFS and OS than those with negative AR. AR may be an independent prognostic marker regardless of ER/PR. DOI: 10.3802/jgo.2018.29.e30, and
  • In section 3.2 – A comment can be entered - Patients with ER/PR positive but ESR1 downregulated stage I HGESS were treated with hormone therapy and recurred at 12 and 36 months after primary resection. ESR1 downregulation is seen in some HGESS that express ER and PR, which raises implications in the utility of endocrine therapy in these patients. DOI: 10.1038/s41379-020-00705-6
  • At Discussions section
    • it will be interesting to mention that recent recognition of HGESS with diverse genotypes suggests that some tumors classified as undifferentiated USs may represent misdiagnosed HGESS.DOI: 10.1097/PAS.0000000000001215
    • Combined therapy - Thus, an aromatase inhibitor or a CDK4/6 inhibitor alone should be considered to treat ER-positive LG-ESS and ER-positive BCOR-related HG-ESS patients with distant metastases, particularly those resistant to endocrine therapy alone. DOI: 10.1126/scitranslmed.aav7171
    • The standard approach to surgical treatment for ESS is hysterectomy and bilateral salpingo-oophorectomy. Despite, studies have not shown clinical benefit to oophorectomy (BSO) at the time of hysterectomy for all patients diagnosed with ESS. Small and large non-randomized series do not suggest any worse outcomes in those patients who retained their ovaries.
  • It will be interesting to show a comparative analysis between studies to evaluate the efficacy of different therapeutic schemes according to overall survival or recurrence rate.
  • For a better understanding of the article, it would be useful to classify the treatment in an early and advanced stage of the US, (DOI: 10.1097/JCMA.0000000000000039)
  • Other studies to add in the references
    • 10.1016/j.critrevonc.2019.08.007,
    • (Lange SS, Novetsky AP, Powell MA. Recent advances in the treatment of sarcomas in gynecology. Discov Med. 2014 Sep;18(98):133-40. PMID: 25227754).

Minor revisions

  • Line 42, the introduction of the last classification is indicated (WHO the fifth edition)
  • Line 73 also in advanced or metastatic disease
  • Line 142 reference
  • Line 249 fist – first
  • Line 328 overall survival abbreviation mention
  • Line 354 12,5% - 12.5%

Author Response

Major revisions:

  • Materials and methods – eligible study design, inclusion, and exclusion criteria

Thanks for this remark, we followed your advice and we added them in the text.

  • Line 195 -I suggest adding the total clinical effective rate to quantify the response to treatment according to the article.DOI: 10.1002/cam4.2044 

Thanks for your suggestion, that improves Our manuscript, and we added it accordingly.

  • Line 250 - (NCCN) guideline recommends aromatase inhibitors as preferred therapeutic regimens for LG-ESS (Abu-Rustum, 2021).

Thanks for your observation, we added this source in our manuscript, with the corresponding reference

  • Lines 305-307 and lines 324-327 --- repetitive mention

Thank you very much for this remark too. We removed one of the two repetitive mentions

  • Line 464 there is not only one prospective phase II trial –
    • George, S., Feng, Y., Manola, J., Nucci, M.R., Butrynski, J.E., Morgan, J.A., et al., 2014. Phase 2 trial of aromatase inhibition with letrozole in patients with uterine leiomyosarcomas expressing estrogen and/or progesterone receptors. Cancer 120,738–743,
    • Friedlander M, Benson C, O'Connell RL, Reed N, Clamp A, Lord R, Millan D, Nottley S, Amant F, Steer C, Anand A, Mileshkin L, Beale P, Banerjee S, Bradshaw N, Kelly C, Carty K, Divers L, Alexander L, Edmondson R. Phase 2 study of anastrozole in patients with estrogen receptor/progesterone receptor-positive recurrent low-grade endometrial stromal sarcomas: The PARAGON trial (ANZGOG 0903). Gynecol Oncol. 2021 Apr;161(1):160-165. DOI: 10.1016/j.ygyno.2021.02.016. Epub 2021 Feb 16. PMID: 33608144.

We apologize for the mistake; we corrected the sentence and added these references in the text, also reporting the results of the recent PARAGON trial.

  • I would like only to suggest introducing a sentence in the Discussions section - Co-expression of ER and/or PR with androgen receptor (AR) was associated with significantly better 5-year DFS and OS than those with negative AR. AR may be an independent prognostic marker regardless of ER/PR. DOI: 10.3802/jgo.2018.29.e30, and

Thanks for your suggestion, we added the sentence to our manuscript.

  • In section 3.2 – A comment can be entered - Patients with ER/PR positive but ESR1 downregulated stage I HGESS were treated with hormone therapy and recurred at 12 and 36 months after primary resection. ESR1 downregulation is seen in some HGESS that express ER and PR, which raises implications in the utility of endocrine therapy in these patients. DOI: 10.1038/s41379-020-00705-6 

Thanks for your observation. The aforementioned paper is about HGESS, so we added this source in our manuscript in section 3.1.2. 

  • At Discussions section
    • it will be interesting to mention that recent recognition of HGESS with diverse genotypes suggests that some tumors classified as undifferentiated USs may represent misdiagnosed HGESS.DOI: 10.1097/PAS.0000000000001215

Thanks for your suggestion, we have added it to our Manuscript.

  • Combined therapy - Combined therapy - Thus, an aromatase inhibitor or a CDK4/6 inhibitor alone should be considered to treat ER-positive LG-ESS and ER-positive BCOR-related HG-ESS patients with distant metastases, particularly those resistant to endocrine therapy alone. DOI: 10.1126/scitranslmed.aav7171

Thank you for pointing out at that interesting published paper, however, We think that those results do not fit the scope of Our review, as they report preclinical results of a study conducted on melanoma PDXs. Since the clinical behavior and availability of treatment modalities are extremely different between melanoma and US, We have decided not to include this reference as it would increase confusion in the reader.

  • The standard approach to surgical treatment for ESS is hysterectomy and bilateral salpingo-oophorectomy. Despite, studies have not shown clinical benefit to oophorectomy (BSO) at the time of hysterectomy for all patients diagnosed with ESS. Small and large non-randomized series do not suggest any worse outcomes in those patients who retained their ovaries.

Thank you for the comment. We reported the controversial role on BSO in ESS in the 3.1.1. Section.

  • It will be interesting to show a comparative analysis between studies to evaluate the efficacy of different therapeutic schemes according to overall survival or recurrence rate.

Thank you for your comment. Even though we acknowledge that it would be interesting to show a direct inter-study comparison, after checking the results of the papers that we included in the review, we found out that most of them report completely different measures of outcome (as in differences in response rates, overall survival, relapse free survival, etc.). Then, it would be impossible to show a direct comparison and to speculate on that. Despite that, we decided to include some sort of inter-study comparison in Our review by showing the different results, that were reported in Table 1 and Table 2. We believe that this might be satisfactory as a reply to your request.

  • For a better understanding of the article, it would be useful to classify the treatment in an early and advanced stage of the US, (DOI: 10.1097/JCMA.0000000000000039)

Thank you for your comment. However, most papers reported in Our review include both patients with early stage and patient with advanced disease treated with endocrine therapy, so accordingly with your suggestion We modified tables classifying studies in Early stage vs Advanced disease.

We think that now Tables are easier to read.

  • Other studies to add in the references 
    • 1016/j.critrevonc.2019.08.007,
    • (Lange SS, Novetsky AP, Powell MA. Recent advances in the treatment of sarcomas in gynecology. Discov Med. 2014 Sep;18(98):133-40. PMID: 25227754).

Thanks for your suggestion, we added it to our manuscript

Minor revisions:

  • Line 42, the introduction of the last classification is indicated (WHO the fifth edition)

Thank you for noticing, we added the edition.

  • Line 73 also in advanced or metastatic disease

Thank you for your comment, we have modified the text rewriting the sentence

  • Line 142 reference 

Thanks for this remark, we have added reference in the mentioned paragraphs.

  • Line 249 fist – first

We apologize for this mistake and we corrected it.

  • Line 328 overall survival abbreviation mention

Thanks for the observation, we text has been modified accordingly

  • Line 354 12,5% - 12.5%

Thanks for the observation, we corrected the typo.

Reviewer 2 Report

The review by Maccaroni et al. summarizes current treatment options for uterine sarcomas.  While all treatment options are discussed, the review focuses on hormonal therapy as a new therapeutic option.  Although the review is interesting, it suffers from many grammatical errors which often make the review difficult to understand.  While I am summarizing some of these issues below, there are mistakes in almost every paragraph.

Grammatical mistakes can be found throughout the manuscript. There are to many to list, but here are just some of the mistakes within the first 25 lines. For example:

  1. Line 17 should be “patients” instead of “patient”.
  2. In line 18, “analyzing” is spelled incorrectly.
  3. In line 19, “in USs” has not been defined. I recommend using “US” as the abbreviation.
  4. Line 22, USs prognosis is still poor should be “are still poor”

Many abbreviations are used but not defined.  For example:

  1. On line 32: LGESS needs to be defined as a low grade endometrial stromal sarcoma
  2. Line 32-33: These terms need to be defined:  uLMS, HGESS, and USS.  What is USS?  Do the authors mean USs?

Lines 55-70 should be rewritten.  This paragraph is wordy and again has many grammatical errors.

Line 81:  Citations should be given as to which retrospective studies were analyzed. 

Line 94:  The authors write “a few reports” but no citations are given.

Line 121:  I am unclear what this means.  “defining specific therapies and their sequences”.  

Line 129 needs to be rewritten:  Although BSO (NEEDS TO BE DEFINED) would be preferred for LGESS, ovaries preservation is possible for young patients with stage I low grade ESS, as it doesn’t promote the risk of recurrence.

Line 305:  It is unclear what the authors are trying to say by this sentence:  Moreover, in a study by Ioffe et al. including 54 women affected by uterine sarcomas, it was found that 100% of uLMS were ER positive, even only 13 (24%) out of 54 patients included had a uLMS. 

Line 310:  “In addiction to that” should read “In addition to that”

Line 444:  This sentence is a run-on and confusing:  Beyond progestins, also AIs are active in patients with LGEES, in particular they seem to achieve a nearly equivalent or even better response as a first-line therapy than as a second-line therapy after the failure of progestins [29,38,39], while data regarding GnRh-a are still to scarce to establish their efficacy in LGESS patients.

Line 464: This sentence is a run-on and confusing.  Only one prospective phase II trials by Slomovitz et al. [63] have been published, in order to evaluate the role of adjuvant letrozole in patients with early stage and positive hormonal receptors uLMS: in this study only 9 patients were enrolled, and even if the progression free rate at 12 and 24 months was 100% for patients receiving letrozole compared to 80% at 12 months and 40% at 24 months for patients in the observation arm, this promising results are based on a sample that is really too scarce to be extended to a wider population.

Author Response

Line 17 should be “patients” instead of “patient”.

We apologize for this mistake and we corrected it 

In line 18, “analyzing” is spelled incorrectly.

We apologize again for this mistake and we have corrected it 

In line 19, “in USs” has not been defined. I recommend using “US” as the abbreviation.

​​Thank you for the comment. Now we have used US as abbreviation in all the manuscript. 

Line 22, USs prognosis is still poor should be “are still poor”

We apologize for this mistake and we corrected it 

Many abbreviations are used but not defined.  For example:

On line 32: LGESS needs to be defined as a low grade endometrial stromal sarcoma

Sorry for the forgotten definition, now we added it before the abbreviation 

Line 32-33: These terms need to be defined:  uLMS, HGESS, and USS.  What is USS?  Do the authors mean USs?

Sorry again, we have defined the abbreviation. USS is undifferentiated uterine sarcoma

Lines 55-70 should be rewritten.  This paragraph is wordy and again has many grammatical errors.

 Thank you for the observation, we rewrote the paragraph.

Line 81:  Citations should be given as to which retrospective studies were analyzed. 

The retrospective studies referred to were those cited by Ferrandina et al mentioned below. (​​Ferrandina, G.; Aristei, C.; Biondetti, P.R.; Cananzi, F.C.M.; Casali, P.; Ciccarone, F.; Colombo, N.; Comandone, A.; Corvo’, R.; De Iaco, P.; et al. Italian Consensus Conference on Management of Uterine Sarcomas on Behalf of S.I.G.O. (Societa’ Italiana Di Ginecologia E Ostetricia). European Journal of Cancer 2020, 139, 149–168. doi:10.1016/j.ejca.2020.08.016). For clarity we mentioned this study in the indicated sentence, too.

Line 94:  The authors write “a few reports” but no citations are given.

The reports referred to were all those mentioned in tables. For clarity we specified in the indicated sentence.

Line 121:  I am unclear what this means.  “defining specific therapies and their sequences”. 

We mean that the lack of large-scale studies, due to the rarity of these diseases, makes it difficult to study them and to clearly define the best sequence of therapies. 

The sentence has been rewritten.

Line 129 needs to be rewritten:  Although BSO (NEEDS TO BE DEFINED) would be preferred for LGESS, ovaries preservation is possible for young patients with stage I low grade ESS, as it doesn’t promote the risk of recurrence.

Thank you for the observation, we have corrected the errors.

Line 305:  It is unclear what the authors are trying to say by this sentence:  Moreover, in a study by Ioffe et al. including 54 women affected by uterine sarcomas, it was found that 100% of uLMS were ER positive, even only 13 (24%) out of 54 patients included had a uLMS. 

Thank you for the observation, the sentence has been rewritten.

Line 310:  “In addiction to that” should read “In addition to that”

We apologize for this mistake and we corrected it 

Line 444:  This sentence is a run-on and confusing:  Beyond progestins, also AIs are active in patients with LGEES, in particular they seem to achieve a nearly equivalent or even better response as a first-line therapy than as a second-line therapy after the failure of progestins [29,38,39], while data regarding GnRh-a are still to scarce to establish their efficacy in LGESS patients.

Thank you for the observation, the sentence has been completely rewritten.

Line 464: This sentence is a run-on and confusing.  Only one prospective phase II trials by Slomovitz et al. [63] have been published, in order to evaluate the role of adjuvant letrozole in patients with early stage and positive hormonal receptors uLMS: in this study only 9 patients were enrolled, and even if the progression free rate at 12 and 24 months was 100% for patients receiving letrozole compared to 80% at 12 months and 40% at 24 months for patients in the observation arm, this promising results are based on a sample that is really too scarce to be extended to a wider population.

Thank you for the observation, the sentence has been completely rewritten.

Globally, the manuscript has been extensively revised, grammatical errors have been corrected and confusing sentences have been rewritten.

Round 2

Reviewer 1 Report

Dear Authors,

With the modifications made clearly and systematically, the present article presents hormone therapy options in uterine sarcomas in a way easily accessible to the reader.

Kindest regards